# Short report: Spatial distribution and growth of sheep farming in Brazilian Amazon

**Andréia Santana Bezerra** [1] *, **Caio Cezar Ferreira de Souza**[2], **Marcos Antônio Souza dos Santos**[3], **Cyntia Meireles Martins**[3], **Maria Lúcia Bahia Lopes**[2], **Alfredo Kingo Oyama Homma**[4], **José de Brito Lourenço Júnior**[1]

**1** Graduate Program in Animal Science (PPGCAN), Institute of Veterinary Medicine, Federal University of Para (UFPA), Federal University of the Amazon (UFRA), Brazilian Agricultural Research Corporation (EMBRAPA), Castanhal, Brazil, **2** Graduate Program in Development and Urban Environment, University of the Amazon (UNAMA), Belém, Brazil, **3** Graduate Program in Agronomy (PGAGRO), Socio-environmental and Water Resources Institute (ISARH), UFRA, Belém, Brazil, **4** Embrapa Eastern Amazon, Belém, Brazil

* andreia.silva@ufpa.br

**Data Availability Statement:** The data used are available from the Instituto Brasileiro de Geografia e Estatística (https://sidra.ibge.gov.br/pesquisa/ppm/quadros/brasil/2020). Data set name:

## Abstract

Sheep farming has grown substantially in the Brazilian Amazon over the previous three decades. This article analyzes the spatial distribution and the dynamics of sheep herd growth using data from the Brazilian Institute of Geography and Statistics—IBGE from 1990 to 2020. The sheep herd growth rates and densities were estimated to assess its dynamics. Maps were then drawn up to show the spatial dynamics of these variables. The highest concentration of herds in the last decade (2010) occurred in Pará, Mato Grosso, and Maranhão states. For each decade there were different growth patterns, but for the entire period (1990 to 2020), there was growth in Mato Grosso, Pará, Maranhão, Tocantins, and Amazonas micro-regions states. The highest density of herd (animals per area) was observed in Maranhão. The potential points for development identified in this study may support strategic planning aimed at strengthening the activity in the region.

## Introduction

The Amazon Forest is a tropical forest located in the northern region of South America. Brazil has the largest part of this biome on the continent, which is equivalent to more than 60% [1]. In the Amazon biome area belonging to Brazil, there is a political delimitation called Legal Amazon, corresponding to the Superintendence for the Development of the Amazon (SUDAM) action area [2]. This region is composed of 52 municipalities in Rondônia, 22 in Acre, 62 in Amazonas, 15 in Roraima, 144 in Pará, 16 in Amapá, 139 in the Tocantins, 141 in Mato Grosso, as well as 181 municipalities in Maranhão State located west of the 44th meridian [3]. Sheep are mainly located in the northeast region of Brazil (14.56 million head), representing 70.59% of the country [4].

However, production systems are extensive and of low technological level [5, 6]. On the other hand, in the South, which has the second-largest herd (3.86 million), constituting 18.75% of total population, sheep production is more specialized, with significant improvement in productivity [7]. Then, there is the Midwest with 4.92% (1.01 million) and the Southeast with 2.99% of the sheep population (616.52 thousand). Finally, the North part, with all the

"Pesquisa da Pecuária Municipal" (table number: 3939). The authors had no special access privileges to the data others would not have.

**Funding:** This study received financial support for publication fee from Pró-Reitoria de Pesquisa e Pós-Graduação (PROPESP/UFPA). Also, the first author received scholarship from the Coordenação de Aperfeiçoamento de Pessoal de Nível Superior (CAPES) – Brasil (Finance Code 001). The funders had no role in study design, data collection and analysis, decision to publish, or preparation of the manuscript.

**Competing interests:** The authors have declared that no competing interests exist.

territory integrating the Brazilian Amazon, has the lowest sheep number (571.26 thousand), representing 2.77% of the country [4]. In this context, the increasing pressure against deforestation in the Amazon region has stimulated the search for animal protein production activities that use minor land extensions. The farming of meat producing sheep could serve as an alternative because it is a medium-sized animal species that need less space than cattle which is historically extensively bred in native and cultivated pastures in the Amazon region [8, 9]. It is noteworthy that sheep farming in Brazil is predominantly performed by family farms, which has relevance in maintaining the traditional population in the region. The fact that this species is a smaller animal facilitates the insertion of small producers in the activity, even for subsistence, preventing the loss of their land to large farms [5, 6, 10]. In addition, these populations use alternative feed sources such as agroindustrial fruit residues and forage adapted to the region (spineless cactus), generating a bioeconomic context for these animals [11, 12].

Moreover, sheep farming has potential for growth instead of increased demand for meat in the domestic market that has been supplied by imported meat, with a quantity of 3.2 thousand tons recorded in the year 2020 [13]. The knowledge about sheep herd growth in Amazon and its distribution would enable public institutions to identify potential growth points. Then, they could elaborate on strategic planning to strengthen the activity in the region and, thus, rationally allocate resources. Therefore, this study was developed to evaluate the sheep herd growth trend and its distribution in the Brazilian Amazon.

## Materials and methods

The research was conducted with data from the Brazilian Institute of Geography and Statistics website from 1990 to 2020. 107 micro-regions of Acre, Amazonas, Roraima, Rondônia, Mato Grosso, Tocantins, Pará, Amapá, Maranhão states were considered, obtaining the herd of the established period (1990–2020) by state and micro-region. The Roraima state was inserted only in 2013 due to lack of information in previous years, and the Maranhão was inserted in its entirety for a complete analysis of the herd dynamics.

The sheep herd density per $km^2$ in each micro-region was calculated by dividing the total existing sheep population by the area in $km^2$.

For producing the herd size and density maps, the categorization by decade was performed by obtaining four maps (1990, 2000, 2010, and 2020 decades) for each variable studied.

For evaluating the sheep herd growth in each legal Amazon state, the growth rates were estimated by regression, using the semi-logarithmic model proposed by Gujarati and Porter [14]. The geometric growth rate (GGR) resulted in four maps. The data were classified into high growth (GGR > 15% per year), moderate growth (8 < GGR ≤ 15% per year), low decrease (-6.49 ≤ GGR ≤ 1 per year) and high decrease (GGR < -6.49% per year).

All maps were made with QGIS software (version 3.16.13).

## Results

### Sheep herd growth

The sheep herd had considerable growth in the legal Amazon from 1990 to 2010 (from 514.9 thousand heads to 1.37 million heads), corresponding to an increase of 62.28%. However, from 2010s to 2020s there was a slight decrease of 2.88%.

Sheep herd was most concentrated in Maranhão and Pará states and over decades began to concentrate in Mato Grosso state (Fig 1).

In the 1990s, the activity was not very representative in the Amazon region (514.95 thousand heads), because the most representative herd numbers were concentrated in some micro regions of the state of Pará (Óbidos, Santarém, Altamira and Ariri) and in some micro regions of the state

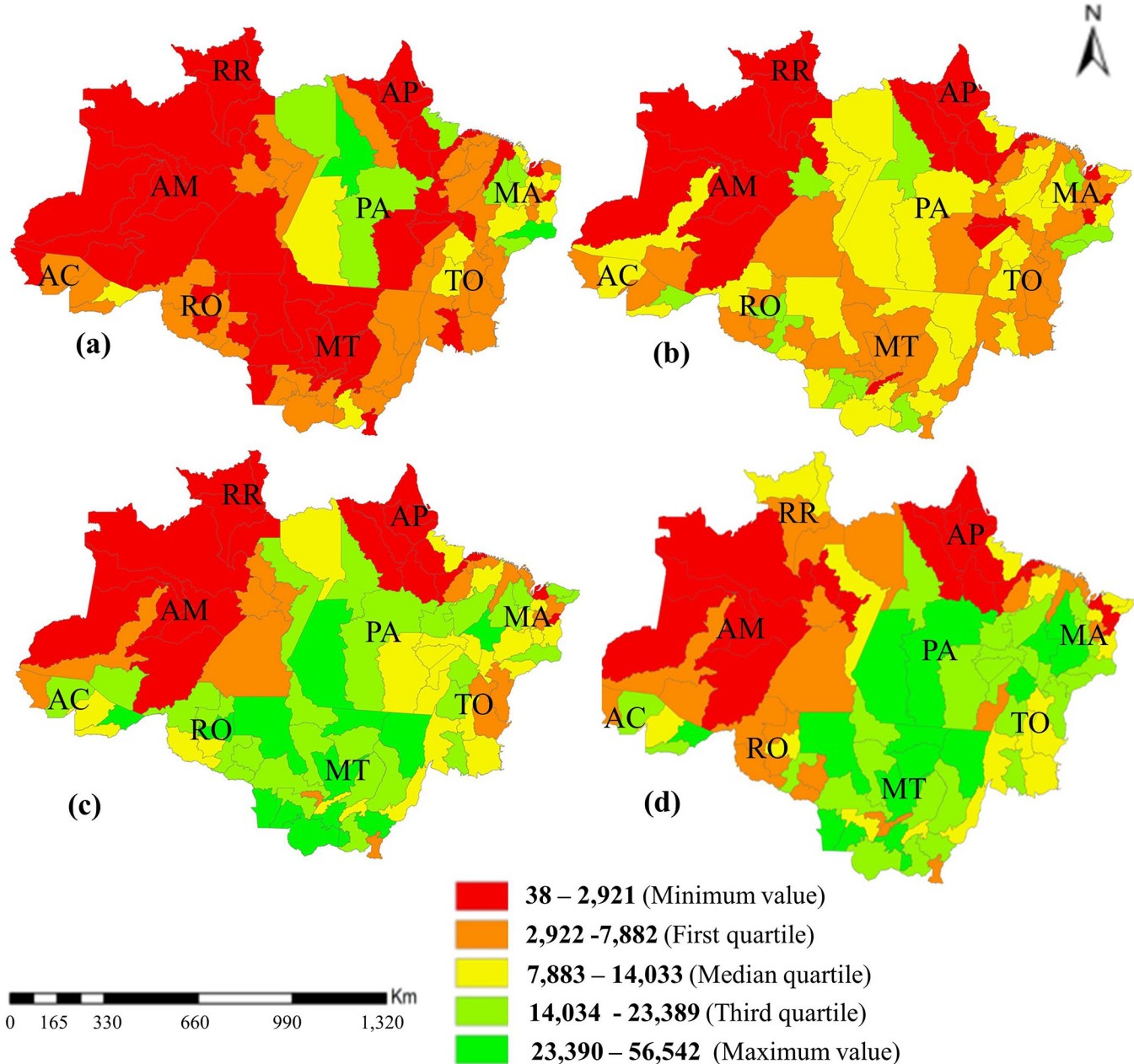

**Fig 1.** Sheep herd distribution in the Legal Amazon (a.1990, b.2000, c.2010, d.2020). RO, Rondônia; AC, Acre; AM, Amazonas; RR, Roraima; PA, Pará; AP, Amapá; TO, Tocantins; MA, Maranhão; MT, Mato Grosso.

of Maranhão (Chapada das Mangabeiras, Chapadas do Alto Itapecuru, Baixo Parnaíba Maranhense, Pindaré and Baixada Maranhense). However, the number of sheep has been growing over the years in the Brazilian Amazon, with the occupation of most micro-regions in Mato Grosso, Acre, those located in the southeast and southwest of Pará, as well as much of those in Rondônia state. As for the 2020s, a larger quantity of animals was concentrated in a part of the Maranhão and Tocantins states. On the other hand, since the 1990s, the activity in the western Amazon states (Acre, Amazonas, Rondonia, and Roraima) is not extensive, with the participation of the four states not reaching 20% of the regional sheep herd in 2020 (Fig 1).

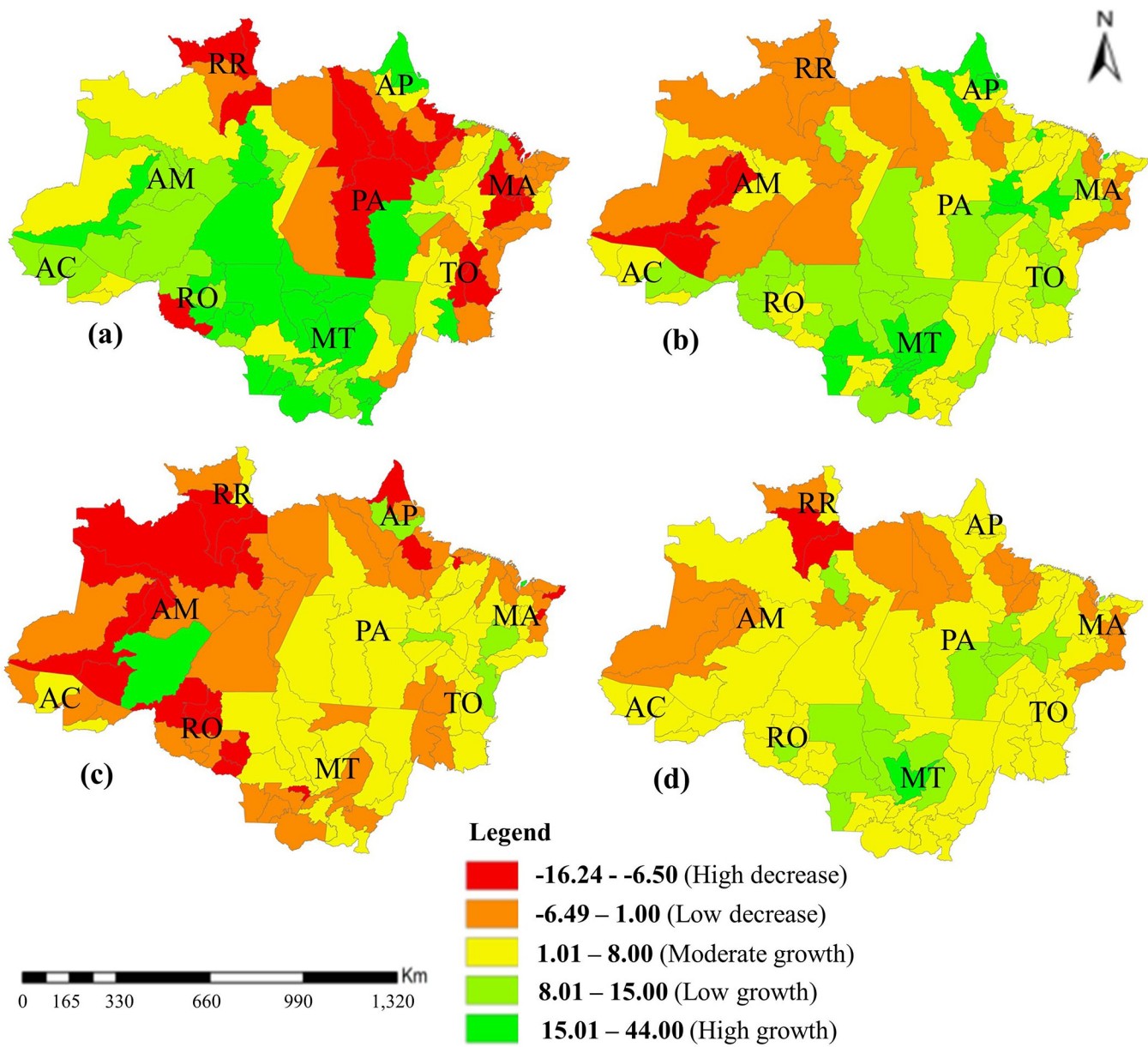

**Fig 2.** Geometric growth rate of sheep herd in Legal Amazon (a.1990-1999, b.2000-2009, c.2010-2020, d.1990-2020). RO, Rondônia; AC, Acre; AM, Amazonas; RR, Roraima; PA, Pará; AP, Amapá; TO, Tocantins; MA, Maranhão; MT, Mato Grosso.

Analyzing the herd geometric growth rate, it is clear that there was a slowdown in the expansion of the herds, showing a decrease in the last decade (2010 to 2020). Regarding the total period (1990 to 2020), a moderate to low growth in eight micro-regions belonging to Mato Grosso state, the majority located in northern Mato Grosso (Aripuanã, Alta Floresta, Parecis, Arinos, Alto Teres Pires, Sinop, Paranatinga) and one in southeastern Mato Grosso (Alto Guaporé) is observed (Fig 2).

There was also an increase in sheep population in most of Southeast Pará, with emphasis on the Tucuruí, São Felix do Xingu, Parauapebas and Marabá microregions; as well as in the imperatriz (Western Maranhão), Bico do Papagaio (Tocantins) and Rio Preto da Eva (Amazonas) microregions. In past decades (1990 to 1999), the growth was more generalized, reaching

a large area of the states of Amazonas, Mato Grosso, Amapá, Acre, and southeastern Pará (Fig 2).

In the entire evaluated period (1990 to 2020), in the Legal Amazon, the geometric growth rate was positive (3.82) but low, classified as low growth (between 1.01 and 8%). The highest rates were found for the Mato Grosso, Acre, and Tocantins states.

## Sheep herd distribution

In general, the density of animals per land area is very low, indicating that this activity is not developed in the Amazon. However, from the 2010s to the 2020s, there was an increase in density in the Maranhão state (Fig 3). It can also be highlighted in Fig 4.

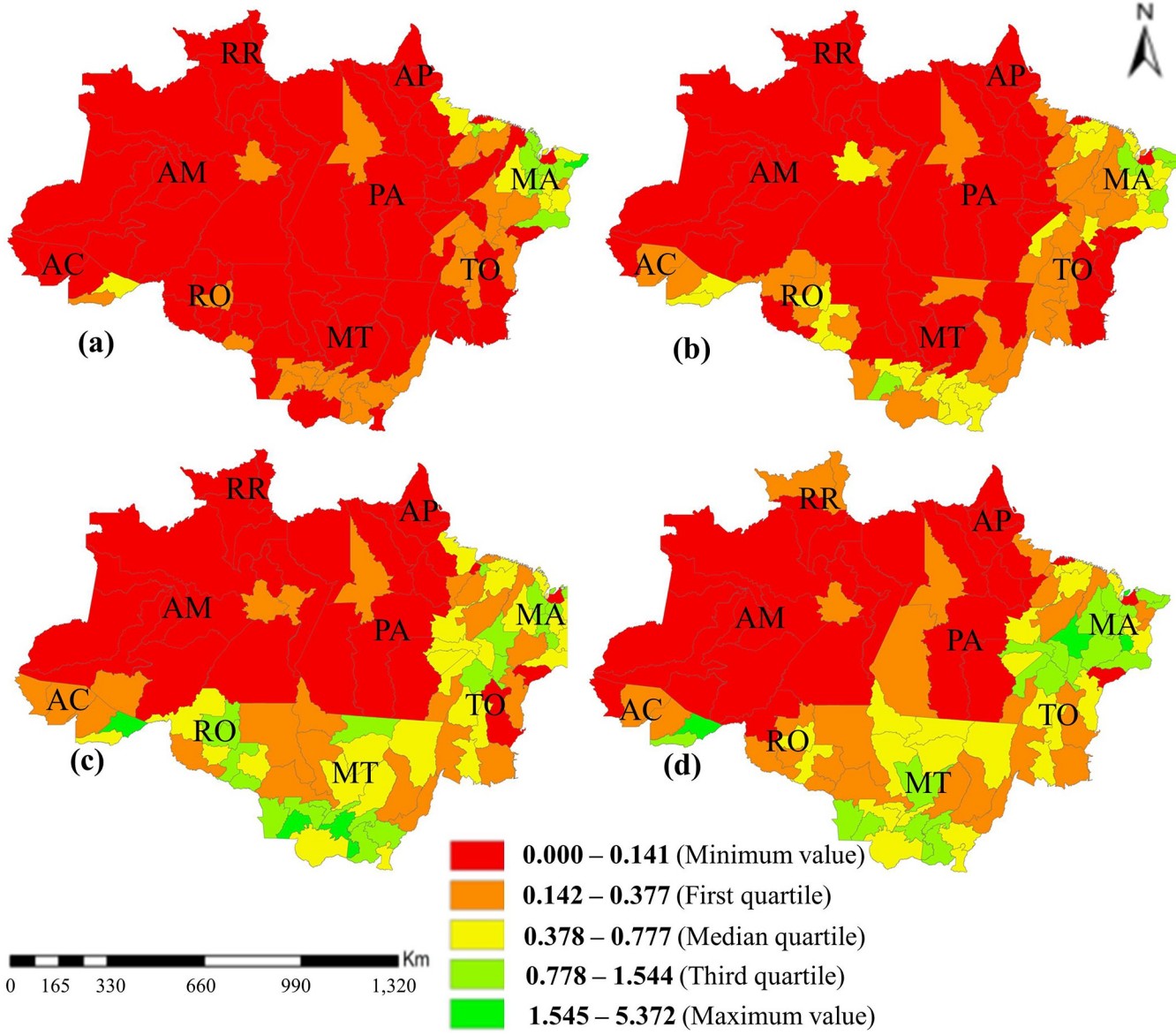

**Fig 3.** Sheep herd density per km$^2$ in Legal Amazon (a.1990, b.2000, c.2010, d.2020). RO, Rondônia; AC, Acre; AM, Amazonas; RR, Roraima; PA, Pará; AP, Amapá; TO, Tocantins; MA, Maranhão; MT, Mato Grosso.

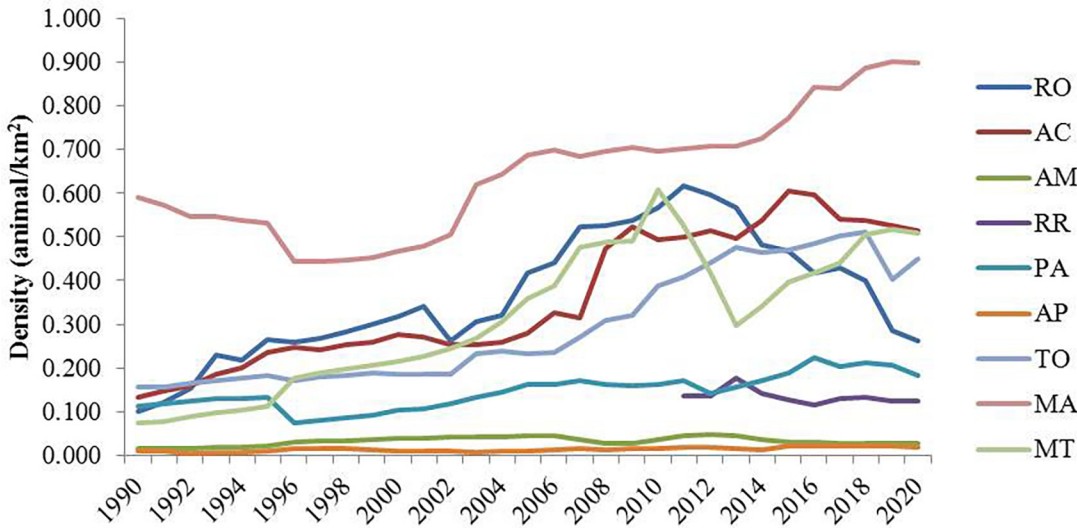

**Fig 4.** Sheep herd density per km² by state (1990–2020). RO, Rondônia; AC, Acre; AM, Amazonas; RR, Roraima; PA, Pará; AP, Amapá; TO, Tocantins; MA, Maranhão; MT, Mato Grosso.

## Discussion

The meat sheep farming scenario in the Brazilian Amazon displayed a potential growth since it showed a growth of 62.26% from 1990 to 2010. This activity could be possibly affected by the pandemic (covid-19) because this crisis affected mainly small producers, and considering that sheep farming in the Brazilian Amazon is predominantly family-based [5, 6, 15–18], sheep farming tended to feel the negative impacts caused by it. The consequences come from sanitary restrictions that triggered the closure of traditional markets (such as open fairs) [19, 20]. In addition, their productions were affected by transportation issues, marketing and storage difficulties, availability of inputs, and access to rural credit [21]. All these variables contributed to an impact on the income of these farmers [22], who may not be able to supply the farm's other activities. However, a more detailed analysis of the impacts of the pandemic on sheep farming in the Amazon region will only be possible in subsequent years, not yet visualized in the analyzed period of this study.

Despite the highest concentration of sheep both in number and density, as well as the growth rate in Mato Grosso, the production of this state is still characterized as being of low technological level [23], with a lack, especially, of appropriate sanitary practices, which decreases productivity. This activity in central Brazil, as well as in all states of the Brazilian Amazon, has been historically developed as secondary to beef cattle farming, and the animals are raised more for subsistence [15, 24].

Moreover, when the producers applied the same handlings as those performed in cattle, they created high productive and income expectations that often were not achieved. As a result, the abandonment of the activity, criticism, and a concept of low profitability in the sector were observed [25]. The socioeconomic profile of farmers in the region with low educational levels and a high degree of illiteracy is also a factor that makes it difficult to obtain information about technological tools and innovations that can contribute to the productive increment [26].

From 1990 to 2010, McManus et al. [27] stated that the high prices of agricultural commodities, environmental restrictions, and high land prices in the Brazilian Midwest region triggered a migration of cattle towards the North of the country. These factors may also have

influenced the increasing adoption of sheep farming, especially in Mato Grosso (Midwestern region of the country), since it would generate animal protein that would occupy a minor extension of land for being a smaller animal [9].

Since most producers secondarily raise sheep in their cattle production [16, 18], this may also have encouraged farmers to adopt sheep farming in the North of the country (especially in the Pará state). It can occur when they moved to the region for better system conditions, seeking greater profitability.

However, the same production model adopted in beef cattle also has been attributed to sheep farming, with extensive rearing and low productivity. Thus, despite being an activity of considerable social and economic importance to the local population, it has low profitability linked to a lack of training for producers and specialized technical assistance [18]. Moreover, sheep farming lacks more investments with the objective to improve reproductive, sanitary and nutritional conditions [16]. Among the management practices, sanitary handling is one of the most challenging in the Brazillian northern region since the general environmental conditions of high rainfall, temperature, and humidity favor infection by gastrointestinal nematodes [28]. The control of these endoparasites is essential because they promote economic losses and damage to the activity, triggered by the mortality of young animals, low weight gain, and increase in feed conversion [29]. Thus, producers need to adopt specific sanitary practices for the soil and climate characteristics of the region to obtain productive success.

Maranhão state has also a significant contribution to sheep production in the Amazon. This state is part of Northeast Brazil, which represents the largest sheep herd [30], accounting for 70.6% of the total sheep in the country [31]. In this region, a semi-arid climate predominates, with long periods of drought and irregular rainfall [32]. Nevertheless, the use of breeds adapted to local conditions has contributed to the growth of production [31]. The rusticity of the species combined with a less demanding zootechnical handling makes them suitable for the system with a low technical level of production. However, productivity is affected by losses in profitability [5, 6], which differs from the production systems adopted in South and Southeastern Brazil that have considerable gains by using specialized breeds and crossbreeds as well as the application of technologies and innovations [7, 33, 34]. The activity in the Northeast is, therefore, characterized by a low technological level, being performed mainly by small producers who practice it in a subsistence format [5, 6]. There is also a great demand for specialized technical assistance and difficulties faced in nutritional management [10]. The fact that this activity is performed by small producers, who have small land extensions, often unsuitable for cultivation, made them use sheep for several decades because they did not have many options [9].

Even though in extensive systems the demand for labor is increased, family farmers seek other ways to supplement income, such as employment outside their property, which helps these families keep their farms running [35]. The participation of small producers in associations and cooperatives makes them more competitive, especially in the midst of crises, such as the one experienced by the pandemic of the covid-19 virus. They provide better infrastructure, elaborate contingency plans, and increase the ability to mitigate the shock of the crisis to their members [20]. They also promote cost reduction and increased profit margin by eliminating intermediaries, adding value to products, and buying and selling in concert [36]. There is also sharing of information [37] to increase production and ease of obtaining rural credit for investments and increased productivity [36]. However, there is a lack of collectivism in some regions of the country, such as the North and Northeast, with the disappointment of many cooperative members to previous experiences, which generates the tendency to individualism [26].

Despite the identification of areas with higher prevalence and potential for sheep herd growth, this activity in the Brazilian Amazon still has a low technological level [5, 6, 15–18, 23], which has restricted its growth. This can be changed with the joining of forces of universities, public agencies, and technical assistance companies in the elaboration of a strategic growth plan to help these family producers, especially in the identified potential points. Thus, bringing to their knowledge both technological tools to increase production and the alternatives for them to obtain rural credit and sell their products.

## Conclusion

The highest prevalence of sheep herds in the last decade (2020) is found almost everywhere in the states of Pará, Mato Grosso, and Maranhão, and a few micro-regions in Acre, Tocantins, and Rondônia, but Maranhão stood out with the highest density of herd (number of animals per area). Through these results, public policies may be encouraged in these potential points to boost the sector in the Amazon region.

There was a slowdown in the expansion of the herd, tending to decline in the last decade. However, when evaluating the entire period (1990 to 2020) it is possible to identify a tendency for the activity to grow in microregions of the state of Mato Grosso (Aripuanã, Alta Floresta, Parecis, Arinos, Alto Teres Pires, Sinop, Paranatinga, Alto Guaporé), Pará (Tucuruí, São Felix do Xingu, Parauapebas and Marabá), Maranhão (Imperatriz), Tocantins (Bico do Papagaio) and Amazonas (Rio Preto da Eva), requiring, however, a look aimed at implementing technological tools to leverage production.

The association of sheep farmers to cooperatives would be a promising alternative capable of boosting productivity growth in this region, given its benefits. Thus, the state incentive is essential and should create methods to clarify to small producers the advantages of this collective organization.

## Author Contributions

**Conceptualization:** Andréia Santana Bezerra, Marcos Antônio Souza dos Santos, José de Brito Lourenço Júnior.

**Data curation:** Andréia Santana Bezerra, Caio Cezar Ferreira de Souza.

**Formal analysis:** Andréia Santana Bezerra, Caio Cezar Ferreira de Souza.

**Funding acquisition:** José de Brito Lourenço Júnior.

**Investigation:** Marcos Antônio Souza dos Santos, Cyntia Meireles Martins, Maria Lúcia Bahia Lopes.

**Methodology:** Andréia Santana Bezerra, Caio Cezar Ferreira de Souza, Marcos Antônio Souza dos Santos.

**Project administration:** José de Brito Lourenço Júnior.

**Software:** Caio Cezar Ferreira de Souza.

**Supervision:** Marcos Antônio Souza dos Santos, Cyntia Meireles Martins, Maria Lúcia Bahia Lopes, Alfredo Kingo Oyama Homma, José de Brito Lourenço Júnior.

**Validation:** Maria Lúcia Bahia Lopes, Alfredo Kingo Oyama Homma.

**Visualization:** Alfredo Kingo Oyama Homma.

**Writing – original draft:** Andréia Santana Bezerra.

**Writing – review & editing:** Caio Cezar Ferreira de Souza, Marcos Antônio Souza dos Santos, Cyntia Meireles Martins, Maria Lúcia Bahia Lopes, Alfredo Kingo Oyama Homma, José de Brito Lourenço Júnior.

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
