## [Editor Report · Decision Letter 0]

2 Jun 2022

PONE-D-22-15390Spatial distribution and growth of sheep farming in Brazilian Amazon PLOS ONE

Dear Dr. Bezerra,

Thank you for submitting your manuscript to PLOS ONE. After careful consideration, we feel that it has merit but does not fully meet PLOS ONE’s publication criteria as it currently stands. Therefore, we invite you to submit a revised version of the manuscript that addresses the points raised during the review process.

We look forward to receiving your revised manuscript.

Kind regards,

Antonio Humberto Hamad Minervino, Ph.D.

Academic Editor

PLOS ONE

Journal Requirements:

2.Thank you for stating the following financial disclosure: 

"This study received financial support for publication fee from Pró-Reitoria de Pesquisa e Pós-Graduação (PROPESP/UFPA). Also, the first author received scholarship from the Coordenação de Aperfeiçoamento de Pessoal de Nível Superior (CAPES) – Brasil (Finance Code 001)."

3.In your Data Availability statement, you have not specified where the minimal data set underlying the results described in your manuscript can be found. PLOS defines a study's minimal data set as the underlying data used to reach the conclusions drawn in the manuscript and any additional data required to replicate the reported study findings in their entirety. All PLOS journals require that the minimal data set be made fully available. For more information about our data policy, please see http://journals.plos.org/plosone/s/data-availability.

4. We note that Figures 1, 2 and 3 in your submission contain [map/satellite] images which may be copyrighted. All PLOS content is published under the Creative Commons Attribution License (CC BY 4.0), which means that the manuscript, images, and Supporting Information files will be freely available online, and any third party is permitted to access, download, copy, distribute, and use these materials in any way, even commercially, with proper attribution. For these reasons, we cannot publish previously copyrighted maps or satellite images created using proprietary data, such as Google software (Google Maps, Street View, and Earth). For more information, see our copyright guidelines: http://journals.plos.org/plosone/s/licenses-and-copyright.

a. You may seek permission from the original copyright holder of Figures 1, 2 and 3 to publish the content specifically under the CC BY 4.0 license.  

Additional Editor Comments:

Dear authors,

Before I can send the manuscript for peer review, some minor issues must be addressed. Please correct all the issues and send back the manuscript.

1- References: there are too many congress citations. According to PLoS One guidelines, this kind of citation is only permitted if they are present online, and must have direct link (https://journals.plos.org/plosone/s/submission-guidelines)

Exemple from PLoS website:

Online articles

Huynen MMTE, Martens P, Hilderlink HBM. The health impacts of globalisation: a conceptual framework. Global Health. 2005;1: 14. Available from: http://www.globalizationandhealth.com/content/1/1/14

2- Figures: Please check figure axis, there is untranslated words. Please check if all figures have the complete information in English, including units and decimal.

3- Parts of the manuscript are in Portuguese:

Fig 3. Densidade do rebanho ovino por km2 132 na Amazônia Legal (a.1990, b.2000, 133 c.2010, d.2020).

4- Figures 1 to 3. It´s not clear what the information in the legend represents. Please clarify the information in the legend. The colors from the figures are different between, please use the same pattern, with the red color representing the higher value. At fig 3 the legend say "higher growth" but the numbers are negative?

5- Conclusion:

Despite the identification of areas with higher prevalence and potential for sheep herd growth, this activity in the Brazilian Amazon still has a low technological level, which has restricted its growth

Considering that you used a data bank, how you can conclude that the sheep herd has low technological level? This paragraph must be moved to discussion with the inclusion or a proper reference that indicate the limited level of technology.
---

## [Author Response · Author response to Decision Letter 0]

17 Jun 2022

Response to reviewers

Journal Requirements:

Response: We have reviewed this and made the missing adjustments mainly in the tables.

2.Thank you for stating the following financial disclosure: 

"This study received financial support for publication fee from Pró-Reitoria de Pesquisa e Pós-Graduação (PROPESP/UFPA). Also, the first author received scholarship from the Coordenação de Aperfeiçoamento de Pessoal de Nível Superior (CAPES) – Brasil (Finance Code 001).” Please state what role the funders took in the study. If the funders had no role, please state: "The funders had no role in study design, data collection and analysis, decision to publish, or preparation of the manuscript.". If this statement is not correct you must amend it as needed. Please include this amended Role of Funder statement in your cover letter; we will change the online submission form on your behalf.

Response: We have included the amended Role of Funder statement in the cover latter.

3.In your Data Availability statement, you have not specified where the minimal data set underlying the results described in your manuscript can be found. PLOS defines a study's minimal data set as the underlying data used to reach the conclusions drawn in the manuscript and any additional data required to replicate the reported study findings in their entirety. All PLOS journals require that the minimal data set be made fully available. For more information about our data policy, please see http://journals.plos.org/plosone/s/data-availability.

Response: We have included the Data Availability statement: “The data used are available from the Instituto Brasileiro de Geografia e Estatística (https://sidra.ibge.gov.br/pesquisa/ppm/quadros/brasil/2020)” in the cover letter.

4. We note that Figures 1, 2 and 3 in your submission contain [map/satellite] images which may be copyrighted. All PLOS content is published under the Creative Commons Attribution License (CC BY 4.0), which means that the manuscript, images, and Supporting Information files will be freely available online, and any third party is permitted to access, download, copy, distribute, and use these materials in any way, even commercially, with proper attribution. For these reasons, we cannot publish previously copyrighted maps or satellite images created using proprietary data, such as Google software (Google Maps, Street View, and Earth). For more information, see our copyright guidelines: http://journals.plos.org/plosone/s/licenses-and-copyright.

Response: The maps included in the study are not copyrighted. They were made by the authors with the assistance of QGIS software (version 3.16.13).

Response: We have revised our reference list as recommended.

Responses to additional Editor Comments

1- References: there are too many congress citations. According to PLoS One guidelines, this kind of citation is only permitted if they are present online, and must have direct link (https://journals.plos.org/plosone/s/submission-guidelines)

Response: We insert the direct link or doi of the articles that were missing to insert. We use Mendeley software.

2- Figures: Please check figure axis, there is untranslated words. Please check if all figures have the complete information in English, including units and decimal.

Response: We checked the figures and made the necessary modifications as recommended.

3- Parts of the manuscript are in Portuguese:

Fig 3. Densidade do rebanho ovino por km2 132 na Amazônia Legal (a.1990, b.2000, 133 c.2010, d.2020).

Response: Thank you for your comment; we have already translated it into English.

4- Figures 1 to 3. It´s not clear what the information in the legend represents. Please clarify the information in the legend. The colors from the figures are different between, please use the same pattern, with the red color representing the higher value. At fig 3 the legend say "higher growth" but the numbers are negative?

Response: Dear Editor, we apologize because this was a mistake. But we have fixed it bying standardizing the colors of the maps with the legends.

5- Conclusion:

Despite the identification of areas with higher prevalence and potential for sheep herd growth, this activity in the Brazilian Amazon still has a low technological level, which has restricted its growth.

Considering that you used a data bank, how you can conclude that the sheep herd has low technological level? This paragraph must be moved to discussion with the inclusion or a proper reference that indicate the limited level of technology.

Response: We moved the paragraph to discussion and included some references that indicate the limited level of technology.

---

## [Decision Letter · Decision Letter 1]

15 Aug 2022

PONE-D-22-15390R1Spatial distribution and growth of sheep farming in Brazilian AmazonPLOS ONE

Dear Dr. Bezerra,

Thank you for submitting your manuscript to PLOS ONE. After careful consideration, we feel that it has merit but does not fully meet PLOS ONE’s publication criteria as it currently stands. Therefore, we invite you to submit a revised version of the manuscript that addresses the points raised during the review process.

ACADEMIC EDITOR:Dear authors, 

The manuscript was revised by experts and it can be accepted for publication after revision. Please take into consideration all the points raised by reviewers when preparing the revised manuscript. I must agree with the reviewer #2, and considering the limited data/analysis, the manuscript should be a short report. Please, change the tittle of the revised manuscript to? Short Report: Spatial distribution and growth of sheep farming in Brazilian Amazon.   

We look forward to receiving your revised manuscript.

Kind regards,

Antonio Humberto Hamad Minervino, Ph.D.

Academic Editor

PLOS ONE

Additional Editor Comments (if provided):

Dear authors,

The manuscript was revised by experts and it can be accepted for publication after revision. Please take into consideration all the points raised by reviewers when preparing the revised manuscript. I must agree with the reviewer #2, and considering the limited data/analysis, the manuscript should be a short report. Please, change the tittle of the revised manuscript to? Short Report: Spatial distribution and growth of sheep farming in Brazilian Amazon.

Reviewers' comments:

Reviewer's Responses to Questions

**Comments to the Author**

1. If the authors have adequately addressed your comments raised in a previous round of review and you feel that this manuscript is now acceptable for publication, you may indicate that here to bypass the “Comments to the Author” section, enter your conflict of interest statement in the “Confidential to Editor” section, and submit your "Accept" recommendation.

Reviewer #1: (No Response)

Reviewer #2: (No Response)

2. Is the manuscript technically sound, and do the data support the conclusions?

Reviewer #1: Partly

Reviewer #2: (No Response)

3. Has the statistical analysis been performed appropriately and rigorously? 

Reviewer #1: N/A

Reviewer #2: (No Response)

4. Have the authors made all data underlying the findings in their manuscript fully available?

Reviewer #1: Yes

Reviewer #2: (No Response)

5. Is the manuscript presented in an intelligible fashion and written in standard English?

Reviewer #1: No

Reviewer #2: (No Response)

6. Review Comments to the Author

Reviewer #1: Manuscript PONE-D-22-15390_R1, entitled “Spatial distribution and growth of sheep farming in Brazilian Amazon”

Recommendation: The above paper is not suitable for publication in its present form.

General comments:

1) The article provides useful information about the spatial distribution and growth of sheep farming in Brazilian Amazon. However, there are a lot of grammar, stylistic and syntax errors. Language needs improvement to become scientific. Please check L57-59, 145-146, 151-152, 171-174, 177-178, 179-180 etc

2) For someone that is not from Brazil, it is difficult to follow the provided maps, without clearly indicating the examined regions. Please use the initials of Figure 4 also in Maps. At the same time, Tables and Maps show the same data. Please choose one way of presentation.

3) Parameters, for example, such as deforestation are associated with sheep farming?

Specific comments:

L20: “previous” instead of “past”

L24: “…dynamics. Maps were then drawn up to show…”

L26: “…Maranhão states. For each decade there were different…”

L29: “observed” instead of “especially”

L41-42: “Sheep are mainly located in the northeast region of Brazil (14.56 million head), representing…”

L42-43: “However, production systems are extensive and of low technological level [5,6].”

L45: “…total population, sheep production is more specialized, with significant improvement in productivity [7].”

L46-47: “…with 2.99% of the sheep population (616.52 thousand). Finally, the North part, with all…”

L48: “sheep number” instead of “herd”

L51: “The farming of meat producing sheep could serve as an alternative because it is a…”

L52-53: “…than cattle which is historically extensively bred in native and cultivated…”

L68: “due to” instead of “for”

L74: “performed” instead of “done”

L82: “with” instead of “in”

L86: “(from 514.9 thousand to 1.37 million heads)”

L94: “incipient”?

L104: “robust”?

L108-109: “…growth rate, it is clear that there was a slowdown in the expansion of the herds, showing a decrease…”

L110: Please delete “one can highlight”

L113: “…Mato Grosso (Alto Guaporé) is observed (Fig 2).”

L114: “There was also an increase in sheep population in most of Southeast Pará…”

L130: “is not developed” instead of “has not occupied much space”

L140-141: “…in the Brazilian Amazon displayed a potential growth since it showed a growth of 62.26% from 1990 to 2010.”

L142: You mean 2010-2020?

L143: “This activity could be possibly affected by the pandemic…”

L148: “method”

L150: “All these variables…”

L155: Area? You mean density?

L158: Please delete “the production system”

L161-164: “Moreover, when the producers applied the same handlings as those performed in cattle, they created high productive and income expectations that often were not achieved. As a result, the abandonment of the activity, criticism, and a concept of low profitability in the sector were observed [22].”

L168: “…MacManus et al. [24] stated that the…”

L183-184: “…investments with the objective to improve reproductive, sanitary and nutritional conditions [13].”

L189-190: Reduction in feed conversion is desirable. You mean increase?

L192: “…states have also a significant contribution to sheep…”

L193: “represents”

L196: Please delete “frequent”

L209: In general, in extensive systems, the demand for labor is increased

L213: “Associations and cooperatives”?

Reviewer #2: I have reviewed the manuscript entitled “Spatial distribution and growth of sheep farming in Brazilian Amazon” written by Andréia Santana Bezerra , Caio Cezar Ferreira de Souza, Marcos Antônio Souza dos Santos, Cyntia Meireles Martins, Maria Lúcia Bahia Lopes, Alfredo Kingo Oyama Homma , and José de Brito Lourenço Júnior and submitted to PLOS ONE as an article. The authors tried to illustrate the growth trend and its distribution of sheep in the Brazilian Amazon. Considering the limited analysis technique they used and the result they have, I suggest the authors to submit the manuscript as research communication/short communication not as a research article.

7. PLOS authors have the option to publish the peer review history of their article (what does this mean?). If published, this will include your full peer review and any attached files.

Reviewer #1: No

Reviewer #2: **Yes: **Samson Leta

---

## [Author Response · Author response to Decision Letter 1]

7 Sep 2022

ACADEMIC EDITOR:

Dear authors, 

The manuscript was revised by experts and it can be accepted for publication after revision. Please take into consideration all the points raised by reviewers when preparing the revised manuscript. I must agree with the reviewer #2, and considering the limited data/analysis, the manuscript should be a short report. Please, change the tittle of the revised manuscript to? Short Report: Spatial distribution and growth of sheep farming in Brazilian Amazon. 

 Response: We have made this modification, as recommended. 

Response to reviewer 1

General comments:

Point 1. The article provides useful information about the spatial distribution and growth of sheep farming in Brazilian Amazon. However, there are a lot of grammar, stylistic and syntax errors. Language needs improvement to become scientific. Please check L57-59, 145-146, 151-152, 171-174, 177-178, 179-180 etc

Response 1: A full English revision was made in the manuscript. The sentences and grammar were revised and improved.

Point 2. For someone that is not from Brazil, it is difficult to follow the provided maps, without clearly indicating the examined regions. Please use the initials of Figure 4 also in Maps. At the same time, Tables and Maps show the same data. Please choose one way of presentation.

Response 2: We indicate the examined regions on maps and provide a legend. 

We have removed tables.

Point 3. Parameters, for example, such as deforestation are associated with sheep farming?

Response 2: Because sheep farming use minor land extensions since meat sheep configures it is a medium-sized animal species that need less space than cattle which is done historically extensively in native and cultivated pastures in the Amazon region. Also, this activity in Brazil is predominantly performed by family farms and the fact that this species is a smaller animal facilitates the insertion of small producers in the activity, even for subsistence, preventing the loss of their land to large farms.

We insert more information about it.

Specific comments:

Point 4. L20: “previous” instead of “past”

Response 4: We have changed it, as recommended.

Point 5. L24: “…dynamics. Maps were then drawn up to show…”

Response 5: We have modified it, as recommended.

Point 6. L26: “…Maranhão states. For each decade there were different…”

Response 6: We have modified it, as recommended.

Point 7. L29: “observed” instead of “especially”

Response 7: We have changed it, as recommended.

Point 8. L41-42: “Sheep are mainly located in the northeast region of Brazil (14.56 million head), representing…”

Response 8: We have modified it, as recommended.

Point 9. L42-43: “However, production systems are extensive and of low technological level [5,6].”

Response 9: We have modified it, as recommended.

Point 10. L45: “…total population, sheep production is more specialized, with significant improvement in productivity [7].”

Response 10: We have modified it, as recommended.

Point 11. L46-47: “…with 2.99% of the sheep population (616.52 thousand). Finally, the North part, with all…”

Response 11: We have modified it, as recommended.

Point 12. L48: “sheep number” instead of “herd”

Response 12: We have changed it, as recommended.

Point 13. L51: “The farming of meat producing sheep could serve as an alternative because it is a…”

Response 13: We have modified it, as recommended.

Point 14. L52-53: “…than cattle which is historically extensively bred in native and cultivated…”

Response 14: We have modified it, as recommended.

Point 15. L68: “due to” instead of “for”

Response 15: We have changed it, as recommended.

Point 16. L74: “performed” instead of “done”

Response 16: We have changed it, as recommended.

Point 17. L82: “with” instead of “in”

Response 17: We have modified it, as recommended.

Point 18. L86: “(from 514.9 thousand to 1.37 million heads)”

Response 18: We have modified it, as recommended.

Point 19. L94: “incipient”?

Response 19: We have changed this word to “not very representative”.

Point 20. L104: “robust”?

Response 20: We mean “extensive”.

Point 21. L108-109: “…growth rate, it is clear that there was a slowdown in the expansion of the herds, showing a decrease…”

Response 21: We have modified it, as recommended.

Point 22. L110: Please delete “one can highlight”

Response 22: We have deleted it, as recommended.

Point 23. L113: “…Mato Grosso (Alto Guaporé) is observed (Fig 2).”

Response 23: We have modified it, as recommended.

Point 24. L114: “There was also an increase in sheep population in most of Southeast Pará…”

Response 24: We have modified it, as recommended.

Point 25. L130: “is not developed” instead of “has not occupied much space”

Response 25: We have modified it, as recommended.

Point 26. L140-141: “…in the Brazilian Amazon displayed a potential growth since it showed a growth of 62.26% from 1990 to 2010.”

Response 26: We have modified it, as recommended.

Point 27. L142: You mean 2010-2020?

Response 27: No, we refer only to the year 2020. However, as we realized that this would not be representative, we decided to remove this value.

Point 28. L143: “This activity could be possibly affected by the pandemic…”

Response 28: We have modified it, as recommended.

Point 29. L148: “method”

Response 29: We have deleted the phase “the predominant methods of selling their products” to make the sentence clearer.

Point 30. L150: “All these variables…”

Response 30: We have modified it, as recommended.

Point 31. L155: Area? You mean density?

Response 31: Yes, we have changed “area” to “density”.

Point 32. L158: Please delete “the production system”

Response 32: We have deleted it.

Point 33. L161-164: “Moreover, when the producers applied the same handlings as those performed in cattle, they created high productive and income expectations that often were not achieved. As a result, the abandonment of the activity, criticism, and a concept of low profitability in the sector were observed [22].”

Response 33: We have modified it, as recommended.

Point 34. L168: “…MacManus et al. [24] stated that the…”

Response 34: We have modified it, as recommended.

Point 35. L183-184: “…investments with the objective to improve reproductive, sanitary and nutritional conditions [13].”

Response 35: We have modified it, as recommended.

Point 36. L189-190: Reduction in feed conversion is desirable. You mean increase?

Response 36: We are sorry; we meant to say “increase”. We have changed “reduction” to “increase”

Point 37. L192: “…states have also a significant contribution to sheep…”

Response 37: We have modified it, as recommended.

We apologize, as we realize that by an oversight we have classified Tocantins state of as being from the Northeast region. However, we have already modified the paragraph leaving only Maranhão state.

Point 38. L193: “represents”

Response 38: We have modified it, as recommended.

Point 39. L196: Please delete “frequent”

Response 39: We have deleted it.

Point 40. L209: In general, in extensive systems, the demand for labor is increased

Response 40: We have modified the text according to your recommendation.

Point 41. L213: “Associations and cooperatives”?

Response 41: We have modified the text to make it clearer.

Response to reviewer 2

Reviewer #2: I have reviewed the manuscript entitled “Spatial distribution and growth of sheep farming in Brazilian Amazon” written by Andréia Santana Bezerra , Caio Cezar Ferreira de Souza, Marcos Antônio Souza dos Santos, Cyntia Meireles Martins, Maria Lúcia Bahia Lopes, Alfredo Kingo Oyama Homma , and José de Brito Lourenço Júnior and submitted to PLOS ONE as an article. The authors tried to illustrate the growth trend and its distribution of sheep in the Brazilian Amazon. Considering the limited analysis technique they used and the result they have, I suggest the authors to submit the manuscript as research communication/short communication not as a research article.

Response: We have made this modification, as recommended.

---

## [Decision Letter · Decision Letter 2]

31 Oct 2022

PONE-D-22-15390R2Short Report: Spatial distribution and growth of sheep farming in Brazilian AmazonPLOS ONE

Dear Dr. Bezerra,

Thank you for submitting your manuscript to PLOS ONE. After careful consideration, we feel that it has merit but does not fully meet PLOS ONE’s publication criteria as it currently stands. Therefore, we invite you to submit a revised version of the manuscript that addresses the points raised during the review process.

The manuscript was reviewed and before it can be accepted for publication, minor revisions are still required. Please see the comments made by reviewer and revise the manuscript accordingly.

We look forward to receiving your revised manuscript.

Kind regards,

Antonio Humberto Hamad Minervino, Ph.D.

Academic Editor

PLOS ONE

Journal Requirements:

Additional Editor Comments:

Dear authors,

The manuscript was reviewed and before it can be accepted for publication, minor revisions are still required. Please see the comments made by reviewer and revise the manuscript accordingly.

Reviewers' comments:

Reviewer's Responses to Questions

**Comments to the Author**

1. If the authors have adequately addressed your comments raised in a previous round of review and you feel that this manuscript is now acceptable for publication, you may indicate that here to bypass the “Comments to the Author” section, enter your conflict of interest statement in the “Confidential to Editor” section, and submit your "Accept" recommendation.

Reviewer #1: All comments have been addressed

Reviewer #2: All comments have been addressed

2. Is the manuscript technically sound, and do the data support the conclusions?

Reviewer #1: Yes

Reviewer #2: Yes

3. Has the statistical analysis been performed appropriately and rigorously? 

Reviewer #1: Yes

Reviewer #2: (No Response)

4. Have the authors made all data underlying the findings in their manuscript fully available?

Reviewer #1: Yes

Reviewer #2: (No Response)

5. Is the manuscript presented in an intelligible fashion and written in standard English?

Reviewer #1: Yes

Reviewer #2: Yes

6. Review Comments to the Author

Reviewer #1: Authors made the majority of the necessary amendments. Some minor points should be corrected before the acceptance of their article.

L40: Please delete "representing" (two times, repetition)

L54: Please delete "e"

L57-59: Please delete (repetition)

L60: "feed sources"

L61: "It inserts"? Please rephrase

L66-67: Please rephrase

L76: "due to" instead of "for"

L168: "...growth rate in Mato Grosso..."

Reviewer #2: My previous comments have been addressed. One minor comment I have pis the type of color used for mapping.

Red color is not a good option to indicate the occurrence of a species like sheep. We often use red color while we map a species which have a detrimental effect.

7. PLOS authors have the option to publish the peer review history of their article (what does this mean?). If published, this will include your full peer review and any attached files.

Reviewer #1: No

Reviewer #2: No

---

## [Author Response · Author response to Decision Letter 2]

4 Nov 2022

Reviewer #1: Authors made the majority of the necessary amendments. Some minor points should be corrected before the acceptance of their article.

Ponit 1. L40: Please delete "representing" (two times, repetition)

Response: We have deleted it.

Ponit 2. L54: Please delete "e"

Response: We have deleted it.

Ponit 3. L57-59: Please delete (repetition)

Response: We have deleted it.

Ponit 4. L60: "feed sources"

Response: We have changed it.

Ponit 5. L61: "It inserts"? Please rephrase

Response: We have modified it to: “…, generating a bioeconomic context for these animals.”

Ponit 6. L66-67: Please rephrase

Response: We have modified it.

Ponit 7. L76: "due to" instead of "for"

Response: We have changed it.

Ponit 8. L168: "...growth rate in Mato Grosso..."

Response: We have modified it.

Reviewer #2: My previous comments have been addressed. One minor comment I have pis the type of color used for mapping.

Red color is not a good option to indicate the occurrence of a species like sheep. We often use red color while we map a species which have a detrimental effect.

Response: Dear reviewer, we appreciate your comment, but we use the colors available in the program that formulated the maps to be able to clearly differentiate higher and lower levels of herd size and growth rate. We use the color red only to indicate the lower values of herd size and growth rate and differentiate them from the higher levels (colors ranging from dark green to yellow).

---

## [Editor Report · Decision Letter 3]

22 Nov 2022

Short Report: Spatial distribution and growth of sheep farming in Brazilian Amazon

PONE-D-22-15390R3

Dear Dr. Bezerra,

We’re pleased to inform you that your manuscript has been judged scientifically suitable for publication and will be formally accepted for publication once it meets all outstanding technical requirements.

Kind regards,

Antonio Humberto Hamad Minervino, Ph.D.

Academic Editor

PLOS ONE

Additional Editor Comments (optional):

Dear authors,

I am glad to inform that your manuscript was satisfactorily revised and now it can be accepted for publication at PLoS One.
---

## [Editor Report · Acceptance letter]

24 Nov 2022

PONE-D-22-15390R3 

Short Report: Spatial distribution and growth of sheep farming in Brazilian Amazon 

Dear Dr. Bezerra:

I'm pleased to inform you that your manuscript has been deemed suitable for publication in PLOS ONE. Congratulations! Your manuscript is now with our production department. 

Kind regards, 

on behalf of

Dr. Antonio Humberto Hamad Minervino 

Academic Editor

PLOS ONE